# “We All Really Need to just Take a Breath”: Composite Narratives of Hospital Doctors’ Well-Being during the COVID-19 Pandemic

**DOI:** 10.3390/ijerph18042051

**Published:** 2021-02-19

**Authors:** Jennifer Creese, John-Paul Byrne, Edel Conway, Elizabeth Barrett, Lucia Prihodova, Niamh Humphries

**Affiliations:** 1Royal College of Physicians of Ireland, Dublin D02 X266, Ireland; johnpaulbyrne@rcpi.ie (J.-P.B.); luciaprihodova@rcpi.ie (L.P.); niamhhumphries@rcpi.ie (N.H.); 2DCU Business School, Dublin City University, Dublin D09 V209, Ireland; edel.conway@dcu.ie; 3School of Medicine, University College Dublin, Dublin D04 V1W8, Ireland; elizabeth.barrett@ucd.ie; 4Children’s University Hospital Temple Street, Dublin D01 F772, Ireland

**Keywords:** well-being, burnout, health professions, hospital doctors, COVID-19, composite narratives

## Abstract

The coronavirus disease 2019 (COVID-19) pandemic poses a challenge to the physical and mental well-being of doctors worldwide. Countries around the world introduced severe social restrictions, and significant changes to health service provision in the first wave of the pandemic to suppress the spread of the virus and prioritize healthcare for those who contracted it. This study interviewed 48 hospital doctors who worked in Ireland during the first wave of the pandemic and investigated their conceptualizations of their own well-being during that time (March–May 2020). Doctors were interviewed via Zoom™ or telephone. Interview transcripts were analyzed using structured thematic analysis. Five composite narratives are presented which have been crafted to illustrate themes and experiences emerging from the data. This study found that despite the risks of contracting COVID-19, many doctors saw some improvements to their physical well-being in the first wave of the pandemic. However, most also experienced a decline in their mental well-being due to anxiety, emotional exhaustion, guilt, isolation and poor support. These findings shed light on doctor well-being during COVID-19, and the ways in which they have been affected by the pandemic, both professionally and personally. The paper concludes by highlighting how doctors’ work life and well-being can be better supported during and after the COVID-19 pandemic.

## 1. Introduction

The onset of the coronavirus disease 2019 (COVID-19) pandemic saw global reporting about frontline health workers, including doctors, battling the virus. Stories and images were shared globally of health workers becoming ill and dying with the virus, as well their physical scars and abrasions from the constant use of personal protective equipment (PPE) to protect them from the virus. Initial reports focused on the importance of safeguarding health workers from the physical threat of COVID-19, and ensuring they had suitable access to PPE to prevent COVID-19 transmission and protect their physical well-being. As the pandemic evolved and the PPE supply chain was strengthened, threats to mental well-being came into focus for health workers; anxieties about safety of self, patients and family, the moral injury of rationing limited care resources, and isolation due to new protocols like social distancing at work and tele-working [1,2,3,4]. While some doctors worked on COVID-19 acute care, doctors working in non-acute care also suffered anxiety, fear and guilt over changes to their normal patient care practices, as routine medical care was reduced or shut down to focus resources on COVID-19 [5,6]. Doctors also suffered the same physical and social restrictions and worries about loved ones as the general public, as a result of stay-at-home orders and lockdowns imposed on whole populations [7,8]. The need to protect the mental and physical well-being of doctors (and equally, other health workers) is critical. This has been recognized by the World Health Organisation: ‘*Our health workforce is exhausted, in countries across our region people are at risk of burning out. We have no COVID-19 response if we do not care for our health-care and essential workers: their needs and well-being must be prioritized*’ [9].

### 1.1. Background: Coronavirus Disease 2019 (COVID-19) in Ireland

At the start of 2020, the Irish health system, while at the time reportedly the fifth highest-spend in the European Union [10], was still beleaguered by years of underinvestment and bed closures which had not been reversed following the previous recession [10]. Ireland’s doctors were already working extremely long hours in public hospitals which were overcrowded and understaffed [11]. In both 2018 and 2019 over 100,000 patients in Irish hospitals were treated without being assigned a hospital bed, meaning that they were treated on a portable trolley or in a chair [12]. Of particular concern pre-pandemic was the limited availability of adequately staffed critical- or intensive-care beds; in 2018 Ireland was found to have an 88% ICU bed occupancy rate, above the European Society of Intensive Care Medicine recommendations of 75%, and only 6 intensive-care unit (ICU) beds per 100,000 people, almost half the European average [13]. Linked to this strain, for many years prior to COVID-19, Ireland’s doctors’ well-being was already reported to be poor, with high levels of emotional exhaustion, burnout and poor work–life balance. For example, Hayes et al. found that one-third of hospital doctors in Ireland were suffering from burnout [14], and Humphries et al. found that 73% of hospital doctors in Ireland had a strained work–life balance and high levels of work-family conflict [15]. In both cases these findings compared unfavorably to findings in studies of doctors in other countries [16,17,18].

Public event cancellations and school closures began in mid-March in an attempt to delay rising levels of COVID-19 [19]. At the recommendation of the National Public Health Emergency Team (NPHET), a national stay-at-home order was implemented on 27 March 2020. This involved moving all but essential workers to work from home, the closure of non-essential retail services, the cancellation of public and sporting events, a restriction on non-essential movement beyond a 2 km radius from the home, reductions in public transport, and physical distancing requirements in supermarkets and takeaway services [19,20,21]. These restrictions began to ease in early summer 2020, when interviews for this study were conducted. Public support for healthcare workers in the first wave of the pandemic was high, and many campaigns were established donating meals for frontline workers, displaying support messages on billboards, and mass scheduled candle-lighting and applause.

In hospitals, major changes were made to enhance capacity to meet projected COVID-19 demands. On 13 March 2020, the national Health Service Executive (HSE) took temporary control of 2000 private hospital beds and 8000 private hospital staff for use as surge capacity for the public health system [21]. Existing public hospital staff were redeployed from many non-acute services to COVID-19 acute services. Almost 400 doctors re-registered with the Irish Medical Council to offer their help to meet demand, coming out of retirement or non-medical careers or returning from overseas [20]. Medical school examinations were brought forward to bring a new cohort of approximately 1000 intern doctors into the workforce early [21]. Many hospitals cancelled, limited or virtually delivered outpatient care services and redeployed staff into emergency, infectious disease and intensive care departments from March–June 2020. Many patients also elected to avoid hospital and did not come in for scheduled appointments or treatments [20,21]. Across all specialties and grades, both private and public, these changes brought swift and significant disruptions to doctors’ working lives.

### 1.2. Doctor Well-Being Defined

Well-being is not a simple concept to neatly define; broadly, it involves individuals’ life and work health and satisfaction and ability to meet challenges and stressors, with physical, mental, social and emotional aspects [22,23,24,25]. When considering well-being in this paper, we employ a theoretical framework based on Walton’s Quality of Work Life (QWL) model, which involves a combination of physical, mental, social and emotional aspects and both the home and work domains [26].

The QWL framework is a construct that deals with ‘*the effect of the workplace on satisfaction with the job, satisfaction in non-work life domains, and satisfaction with overall life, personal happiness, and subjective wellbeing*’ [27]. It is comprised of eight conceptual categories:adequate and fair compensation;safe and healthy working conditions;opportunities to develop and use skills and capacities;future opportunities for growth;social integration;organisational constitutionalism in the workplace;recognition of total life space; andsocial relevance of work.

The pre-eminence of different categories will vary amongst individuals with different work and life circumstances, but all are important in some measure for well-being.

There are many consequences of poor workplace well-being: for organisations, lost production, higher occupational health costs, and potential legal costs; and for individuals, increased physical danger, poor workplace relationships, family conflict, depression, anxiety and burnout [22]. Of these, wellbeing and burnout have traditionally gone hand-in-hand in medical literature [24], possibly due to the high prevalence of burnout among healthcare professionals [28]. Burnout is officially classified in the World Health Organisation’s International Classification of Diseases (ICD-11) as an employment-associated disease, ‘*resulting from chronic workplace stress that has not been successfully managed*’ [29]. It is typified by either feelings of depletion and exhaustion, feelings of negativity and cynicism related to one’s job (emotional exhaustion), increased mental distance from one’s job (depersonalization), or a sense of ineffectiveness and lack of accomplishment (low personal achievement). In healthcare, physician burnout has been shown to increase medical errors, hasten exit from the health system or profession, and lead to negative physical coping mechanisms for stress, such as alcohol and drug use, and, in extreme cases, to self-harm and suicide—the latter occurring at a higher rate amongst doctors than in the general public [28,30,31]. While Scaria et al. [24] argue that burnout and well-being are not exact opposites (the opposite of well-being they see as distress, which then may cause burnout), the two factors can operate in synergy; improving well-being on an individual level may boost resilience to avoid burnout, and improving the broader working conditions that lead to burnout may allow individuals to boost their well-being [32].

Despite burnout being understood as a syndrome suffered by individuals, and often treated as such in the workplace, the literature has only recently begun to consider burnout as a product of a workplace environment, rather than of the individual employee [33,34]. While stress is naturally high in caring professions [28,31], workplace programs to help healthcare workers manage that stress are often focused on the individual, e.g., personal mindfulness and resilience training, rather than on the workplace, e.g., improving working conditions, reducing under-staffing or work overload. Programs that focus on doctors’ individual well-being responses and responsibilities have been found to be less successful, long term, than interventions focusing on changing the organisational environment to reduce stressors [14]. Informal supports in workers’ personal lives, such as friends and family, have been shown to protect against burnout and reinforce resilience and coping [31,35]. However, poor work–life balance and work–life conflict, which is widespread among hospital doctors in Ireland [14,15], often mean that doctors have insufficient time outside work to access these informal supports, thereby creating further stress [33]. Colleagues and collegiality may provide some informal protections against stress [31,36,37], although perhaps not if these colleagues are suffering the same stress themselves. Strong institutional programs of well-being that include reflection, intervention and support have been shown both to directly reduce psychological strain, and have a secondary benefit of strengthening team bonds and transferring tools to participants’ home lives, which provide further informal support for well-being [38,39]. Doctors with positive well-being, maintained by good supports at work and at home, can contribute to building an environment where patient care quality is high, costs are well-managed and staff turnover is low [28]. These are strong incentives for health systems to understand the impact of COVID-19 and support doctors’ well-being.

This paper contributes to this literature by considering Ireland’s hospital doctors’ conceptualizations of their own mental and physical well-being during the first wave of the COVID-19 pandemic in Ireland. It will explore thematically how the doctors interviewed (*n* = 48) saw their well-being change during this period, both at work and at home, and will consider ways in which the health system can better support the physical and mental well-being of its health workforce, including hospital doctors.

## 2. Materials and Methods

Qualitative, semi-structured interviews were conducted in June–July 2020 with 48 hospital doctors who had worked in Ireland during the first wave of COVID-19 (March to May 2020). All respondent doctors were hospital-based rather than community-based, and all doctors were Irish citizens. Depending on their specialty, some doctors worked directly with acute patients suffering from COVID-19 (e.g., emergency medicine, public health, respiratory), while others worked in non-acute care but had their work practices and conditions impacted by COVID risks, restrictions and policies. Potential respondents made contact in response to a call for participation circulated on the Twitter social media platform. Participants represent a convenience sample of hospital doctors who responded to the call for interviews. Ethical approval for the research was granted by the Research Ethics Committee of the first author’s institution. The 48 interview participants were a mix of genders, specialties and professional grades, and had different life situations and responsibilities (see Table 1).

All interviews were conducted via Zoom™ or by telephone; informed consent was obtained prior to each interview through an online form. This enabled the research to comply with social distancing requirements, thus protecting the physical health of interviewers and interviewees. Virtual interviews also enabled participants to express any vulnerabilities, emotions and private experiences in their own safe and comfortable space [40]. Interviews lasted an average of 45 min (ranging from 23 min to 93 min). The interview schedule was arranged around seven themes jointly developed by the authors, in line with the specialized research interests comprising the broad scope of the research project. Interviews discussed respondents’ experiences of working as a doctor pre-pandemic, during the pandemic and their plans for the future. This paper focuses on responses to the question ‘*how has the pandemic impacted on your well-being?*’. Interviews were recorded and then transcribed by a third-party service, were de-identified by interviewers and forwarded to interview participants for further de-identification and approval. Structured thematic data analysis was conducted [41], using MaxQDA software. The total body of interview data on doctors’ experiences of COVID-19 was initially analyzed using the framework method [42], with data coded into seven deductive categories jointly developed by the authors in line with the themes of the interviews. Data within the initial code of ‘life and well-being in COVID-19′ were then broken down into sub-codes aligning the QWL framework [26].

These themed findings are presented in this paper in the form of five composite narratives, drawn together from shared themes and experiences found within the full dataset of 48 interviews. Composite narratives in qualitative research involve the use of data from several interviews to tell a story framed as that of a single individual. Composite narratives have been used in qualitative sociology to share the voices of groups such as asylum seekers, politicians, and parolees, who have unique lived experiences and human stories behind the public rhetoric about them, but who may be vulnerable should their individual identities be disclosed [43,44,45]. The doctors interviewed in this study are similarly positioned: they have their own human stories beyond the public rhetoric of the ‘healthcare hero’ [46], but can be reluctant to voice concerns about working conditions in an identifiable way [11]. Composite narratives provide readers with a rich way of connecting to and understanding their experiences while protecting participant anonymity by creating stories from the parts of multiple lives to render individuals unidentifiable [43]. By developing narratives, the authors have generated stories to help readers to better understand the experiences of frontline health workers during the COVID-19 pandemic. The narratives are intended as an effective way ‘of summarizing key issues and concepts simply, quickly and effectively’ [47] and are used in order to enhance the policy impact of this research. Narratives were crafted out of coded data, scaffolded by the elements of the QWL framework, to incorporate a broad range of the individual experiences, realities and philosophies of respondent doctors in a format that met the ‘simply, quickly, effectively’ goal yet retained a human face.

## 3. Results

Five doctors’ narratives are presented below to convey the findings from the data. All names used are pseudonyms, in accordance with the guidelines of the International Sociological Association [48].

### 3.1. Sean

Sean is an intern in a busy city hospital. The COVID-19 pandemic began in Ireland when Sean was in the final months of medical school, and he was fast-tracked into an intern post to help the pandemic efforts. Instead of the usual hospital-wide intern experience, infection control protocols meant Sean was posted to one ward without rotation for the first three months of the pandemic

Sean found his physical health improved during the early pandemic as he found time to take up running as an outlet for work stress. ‘*Over the last two months I’ve run probably more than I’ve ever run. I needed to do something to debrief or to distract myself or to relax, so I was more mindful of doing things like going for a run than I would be in normal times.*’ Sean caught a common cold during this period, and although he tested negative for COVID-19, hospital policy meant he was required to take mandatory sick leave while awaiting test results, allowing him some time to recover before returning to work.

Early in the pandemic, Sean felt adrenaline-charged about working on the hospital frontlines with his fellow interns: ‘*People were energized, they were on an adrenaline high and that kept a lot of people going.*’ However, he noticed that as the months wore on, ‘*there is a weariness on everybody now. We’ve come down from the adrenaline rush of COVID, we all really need to just take a breath, recuperate, regain our energy.*’ He is disappointed at still being ward-based, and is worried that he is missing the opportunity to build relationships for his future career and training.

At home, Sean found lockdown mentally draining with little to do outside of work: ‘*Just the thought of not being able to do anything at the weekend was going to kill me. There was no structure of a week. The mental health that is supported by that framework dissolved.*’ He began volunteering for extra shifts at work to fill the time: ‘*I was just bored at home… and I was actually enjoying being at work*’. Sean lives with his father, who has had a chronic underlying health issue, which has left Sean very concerned he would pass on the virus to him. Sean and his father ‘*divided the house front and back, so although we were living in the same house, we didn’t actually have any contact with each other.*’ Even when restrictions eased and small gatherings were allowed, several of Sean’s friends asked him to avoid group events for fear he would spread the virus: ‘*I was disinvited from stuff, because I was physically working on the front lines, a number of people asked me not to go to their house.*’

One thing that helped Sean was his hospital’s staff support program, which offered regular scheduled mindfulness sessions and active communication from hospital-based staff psychologists. His workplace also offered free car parking for doctors, changing what would normally be an hour-long public transport commute to a 10-min drive. *‘There’s just been really nice, positive initiatives like that around general staff support’*. However, the changes to his living arrangements at home, while crucial to protect his father from COVID-19, lost him a valuable source of support. Maintaining physical distance from his father made it hard for Sean to talk through and decompress from the pressures of his workday as he used to, and he felt isolated in his own home, as well as anxious that he might slip up and pass the virus on to his father.

### 3.2. Bridget

Bridget is a registrar working on a non-COVID ward, with older patients who are at high risk of severe complications should they contract the virus. Bridget’s department had rigorous patient and staff testing guidelines, and her duties involved developing her department’s COVID-19 safety guidelines and communicating them to both junior and senior colleagues. Bridget lives in a large city with a non-medical partner who worked from home throughout lockdown.

Physically, Bridget found her health improved during the lockdown, particularly her diet. She and her partner had more free time to prepare meals at home, rather than eat takeaway food. She reflected that before COVID-19, *‘The day was just so intense that you were just absolutely exhausted in the evening. I put on about 15 kilos in a year of weight just from stress eating’*. But during the lockdown, *‘I have way more energy in the evenings. I can cook, I never used to cook, I was just eating takeaways all the time. I’ve been losing weight since COVID started and I’ve been cooking for myself.’*

Mentally, Bridget found that working during the pandemic made her stressed, anxious, and guilty. She was tasked with helping to administer daily COVID-19 tests to elderly patients, swabbing their noses and throats. Most of these patients suffered from dementia and were very frail, and swabbing them was a stressful process for both patient and doctor: *‘You feel like a monster, an elderly person begging you not to swab them. Really, it was a little harrowing.’* Her COVID-19 policy design role often placed her in situations of interpersonal conflict at work. *‘Everything became my problem, this junior person trying to tell someone in their 50s “you have to go to this ward, this is where you’re being placed, or this is why we’re doing this”.’* Bridget was highly anxious about these duties causing professional backlash against her and affecting her future career trajectory.

At home, Bridget keenly felt the loss of her social life: ‘*Just simple things… not meeting your friends, not getting your date night out. Not going to the gym anymore, all those types of things. That was really tough, because normally when work is difficult, you have an outlet*’. Bridget enjoyed going to work as a replacement social outlet and often volunteered to provide backup and cover for colleagues on her own days off. ‘*I think it’s almost made me see it a bit more positively in a way because I can connect with colleagues.*’ However, this brought guilt at home, as her partner was unable to do the same: ‘*He’s going stir-crazy… couldn’t get out and do anything. While I’m leaving during the day to go out and have what, essentially, was my normal routine.*’

While Bridget felt safe from the virus through her workplace’s strong, public health-informed policies, she felt that hospital administrators had neglected to implement policies and practices to protect and support staff mental well-being. Bridget’s workplace offered the staff a series of wellness webinars, but she found them unsatisfying: *‘The emotional exploration of how people are is very artificial. It would be nice if someone just asked me was I getting to see anyone or having any contact with anyone.’*

### 3.3. Shannon

Shannon is a specialist registrar in a city hospital. She worked in a busy acute department, where urgent initial care would be provided to patients brought in with severe COVID-19 infections. Large numbers of staff were redeployed to cope with any potential surge in patient numbers. Shannon has two young children, and a husband who is a medical consultant.

Shannon contracted COVID-19 in June, was tested as soon as she showed symptoms, and was immediately placed on two weeks’ isolation leave. Her symptoms were minor, but hospital policy required her to remain home. While she felt guilty for leaving her team, she appreciated this mandatory leave as in the past she had felt pressure to return to work as soon as possible when sick. *‘In normal life, we would have gotten back to work just feeling a bit rubbish, but this enforced rest. I went back to work feeling well, which is really unusual after being sick.’* Her physical fitness declined during the pandemic as she was unable to continue her swimming exercise routine when the swimming pools closed during the lockdown, and *‘I turned to chocolate and wine in the early days to cope… I’ve put on a lot of weight, and I’m really conscious of that, actually.’*

Shannon found the extra staffing in her department during COVID-19 greatly eased her roster and allowed her to take lunch and rest breaks and finish work on time, which under normal circumstances rarely occurred: *‘I’m not used to having a day where you can go for a coffee after your ward rounds and chill out for a half an hour. I’ve never had an experience working in an Irish hospital like this.’* This reduced workload boosted her mood significantly, although she was concerned that this did not feel like *‘battling’* against the pandemic, and was anxious that there might be tasks she was missing. She also felt anxious about what was happening to patients whose outpatient appointments were cancelled, or who were avoiding attending hospital for minor complaints, deferring attendance until they became chronically unwell: *‘The people coming in though are definitely a lot sicker than previously. I mean there’s a lot less admissions, but the admissions are a lot more scary.’*

At home, Shannon’s pandemic experience was extremely stressful. The family’s childminder quit early in the pandemic, claiming the family was *‘too high risk’*. Left without childcare, Shannon and her husband sent their two young children to live with relatives. The couple did not see their children for several weeks. Even with no children in the house, Shannon found decompressing from her work during the pandemic impossible at home. *‘All you were thinking about when you came home was the pandemic, and then that’s all you were thinking about at work. There was nothing except COVID-19 and when you turned on the TV it felt like that was the only thing on the news.’* The experience of family separation, coupled with contracting the virus, has left Shannon questioning her future in the medical profession: *‘I’m definitely 100% going to look to actually changing the way I do things now, to be a good mother. That has to be my priority.’*

Communication about both COVID-19 safety guidelines and staff support was poor in Shannon’s workplace. *‘There was a huge amount of chopping and changing. The guidelines were changing so often.’* This increased her sense of anxiety and isolation. Her workplace offered no well-being supports or sessions for staff beyond generic flyers, so she ended up seeking private therapy sessions out-of-pocket to help her deal with the effects of working through the pandemic and to help her cope with the separation from her children. *‘Our well-being is always put on the back burner. It’s really frustrating, and upsetting’*.

### 3.4. Fiona

Fiona is a consultant working in a regional hospital, managing a team of junior doctors in a non-acute medical ward and clinic service. Many of Fiona’s junior doctors were reassigned to COVID services during the pandemic, disrupting their training. Fiona has a husband in a non-medical career and two children who required home-schooling during the lockdown. She also has older parents, one with a chronic health condition, who live nearby.

Physically, Fiona’s health improved a little during the pandemic; without the rushed morning commute, she and her family had more time for exercise and a nutritious breakfast. *‘I got up every morning and I did a bit of exercise, I had my breakfast. I’ve been doing the Joe Wicks [online exercise classes] with the kids.’* However, anxiety interrupted her sleep, which affected both her physical and mental health: *‘I’d say there was a good two to three weeks where I didn’t get any sleep at all just from worrying.’*

In the early days of the pandemic, Fiona was extremely worried about what might be coming to Ireland, seeing reports in the media and from colleagues in Italy and Spain. She knew from years of experience that her own hospital was already thinly stretched. *‘The uncertainty was the worst part, the worry that we wouldn’t be able to cope with it.’* As the first wave of COVID-19 passed without overwhelming the system, she began to worry about a potential, more severe second wave of COVID-19 alongside the usual winter influenza cases. *‘We know that we’re still vulnerable and that’s kind of a stress for everybody’*. As well as her ward-based work, Fiona normally sees many patients in clinics, but has changed these to a telehealth model to provide patient care; she finds this frustrating as she struggles with language barriers and the lack of non-verbal cues from her patients, and she feels isolated sitting alone in an office on the telephone for clinic hours. Fiona has also felt anxious and guilty that she has not been able to adequately supervise her trainees. She felt loneliness at missing them, and anxiety and guilt because she might see them going in and out of the hospital, but she was not able to interact with them. *‘There’s just that feeling that you shouldn’t be doing that, unless it’s somebody you directly work with.’*

The stay-at-home order meant Fiona’s children were at home on weekends and evenings, rather than out at school, playing sport or out with friends. The family could spend more time together, but at the same time, they found it difficult to get time and space to themselves. Despite the pressure of the hospital, going into work gave Fiona some time alone away from her children and out of the house. But this escape also brought feelings of guilt, *‘because I was able to go places and see people and my family couldn’t.’* She also worried about passing the virus on to her children and husband, and had a complex after-work *‘ritual’* designed to prevent contamination*: ‘I would go to work, change into my scrubs, change out of my scrubs at work, come home, the kids weren’t allowed to touch me, everything went in the wash, I went and had a shower, and only then I was able to interact with the kids.’* On the insistence of her sister, she stopped making care visits to her parents, to avoid the potential risk of passing on COVID-19 to them. She managed to arrange alternative care arrangements, but remained anxious for their safety, and felt resentful that she was unable to care for them, or even see them.

Fiona runs a tight-knit team, who normally all actively support one another’s well-being at work. Losing her trainees to redeployment has meant a loss of that support network for them all. *‘There has been no debrief, there has been no team get-togethers, no breaking of bread, no hugs.’* Her hospital offered no specific well-being support for staff; her professional body offers some online resource sheets, but Fiona found they did not deliver the kind of group-based collaborative support she finds most helpful. She instituted regular Zoom debriefs with her former trainees on her own, *‘just to try to address some of the emotional concerns that we had.’* She also arranged for a catering company to provide daily healthy meals for her ward team during the first wave, *‘in an attempt to give them at least one meal a day that was okay’.*

### 3.5. Aidan

Aidan is a senior house officer (SHO) in a regional hospital. At the start of the pandemic, he worked in a frontline service department which quickly became busy with patients presenting with acute COVID-19. Although he was scheduled to rotate back to a city hospital for further training, his rotation was cancelled at short notice as staff were reassigned to COVID services.

Working in acute COVID-19 care was stressful for Aidan. His physical health suffered due to the stress he was under working on the COVID service: *‘Stress eating became a massive thing. Drinking half a bottle of wine a night.’* He found working while wearing extensive PPE *‘suffocating’*, and it caused him to break out in rashes. *‘Everybody was having all of these horrible skin reactions and horrible dermatitis of all types.’*

Aidan contracted coronavirus in late March but struggled to arrange a COVID-19 test through his workplace occupational health department. *‘They’d only just started up staff swabbing, you kind of had to beg them. I had to send a lot of vicious emails.’* By the time he finally got his swab, isolated and tested positive, another SHO colleague had become ill, and his supervising registrar and specialist registrar were ordered into isolation: *‘I was like a sniper, essentially, just sliced everyone and they were down to half the team.’* His flu-like physical symptoms lasted for a week, and he *‘came out of COVID a little bit deconditioned’*.

Although he likely acquired COVID-19 at work, Aidan felt very guilty for contracting the virus and potentially for infecting his colleagues, especially senior colleagues: *‘taking out an intern didn’t really matter but taking out a med reg was a big deal. Or even worse, a consultant’*. Aidan’s consultant was in an older age group, and Aidan became anxious that his infection might make his consultant critically ill. His mental well-being deteriorated so much during this period that he ended up breaking down and seeking stress leave from work: *‘I was crying all the time, stressed from the job and I felt physically sick’.* He was eventually reassigned to a non-COVID-19 service. He attributes his burnout and the need for stress leave directly to being assigned to the acute COVID-19 service for months without a break.

At home, Aidan lives alone in a studio apartment, and when he contracted COVID-19 he was worried that nobody would be able to care for him if his condition worsened. Aidan felt the lockdown experience to be a lonely and isolating one: *‘Everyone I work with has someone to go home to. I was literally just like home, work, home, work… it was hard.’* His social support networks outside work shut down: *‘My non-medical friends wouldn’t even meet me for a coffee with two meters apart because I was at risk’*. During the lockdown, when regional travel was prohibited, he was unable to travel home to his parents, as he normally would do. His parents and family generally provide important support and help him to deal with work pressures.

Aidan tried to avail of the workplace well-being supports emailed to all hospital staff, but found them too unstructured and impersonal; *“Just ‘here’s a number. You might get through to someone or send a text.’”* Eventually, in an informal discussion with his supervising consultant, Aidan admitted he felt close to burning out. She liaised with hospital administration to initiate the stress leave policy for Aidan, but this process was stressful and time-consuming for them both, involving multiple phone calls back-and-forth. Although happy in his relocated role, Aidan reflected that the whole experience of trying to seek support while working in a COVID-facing department was extremely stressful for him and his support network: *‘You’d be kidding yourself if you said that type of experience is good for your well-being’*.

## 4. Discussion

Our study illustrates the range and complexity of issues affecting the physical and mental well-being of hospital doctors during the first wave of the COVID-19 pandemic in Ireland. In terms of doctors’ quality of working life [26], it highlights how the pandemic impacted on their ability to experience safe and healthy working conditions; their opportunity to develop and use their skills and capacities; their social integration; and the recognition of their total life space. Table 2 aligns QWL framework categories with data in the findings/narratives and themes which will be discussed in further detail. The experiences described underscore the need for doctors to have sufficient time away from work to protect their own health and well-being and also the need for appropriate space (physical and metaphorical) to make use of this time, which was often difficult during the first wave of COVID-19. They draw attention to the range of emotions associated with providing patient care during a global pandemic, and how negative emotional stressors might be buffered. They underline the multifaceted human needs of doctors negotiating work–life balance and work–family conflict issues which were intensified during the pandemic. Finally, they examine how psychological and practical supports might better address doctors’ needs after the first wave of the COVID-19 pandemic.

### 4.1. Time and Space for Well-Being

One of the most important factors associated with improving health and well-being is time: ‘*time to think, time to meet with peers, time to reflect… and time to care for oneself and have life outside work*’. In terms of their Quality of Work Life [26], given adequate time and space, doctors could work in safe and healthy working conditions, maintain their social integration and take care of needs across their total life space. Providing this time and space for doctors can also lead to improved patient safety, and prevent the transmission of COVID-19.

Changes to work practices due to COVID-19 had a great effect on doctors’ time. Many of the junior doctors interviewed perceived an improvement in medical staffing levels during the first wave of the pandemic (partly as a result of COVID-related redeployment of staff) which eased workload and time pressures at work [49]. Some work such as administrative tasks and meetings which previously needed to be done physically in the hospital was changed to be undertaken at home during the lockdown, which also allowed doctors to spend more time at home. This enabled respondent doctors to achieve a better work–life balance, and to improve their diets and exercise levels (as Sean, Bridget and Fiona did), similar to findings from a survey of Malaysian doctors [3]. Evidence suggests that exercise levels have increased across the general population globally during COVID-19 lockdowns [50], and doctors may be following this general trend. Other respondents (Bridget and Fiona) found that the pandemic enabled them to leave work on time which allowed more time to enjoy with family, or to spend relaxing, for better recovery of mental and emotional health.

Occupational and public health guidelines mandated sick leave for doctors who contracted COVID-19 or were close contacts of a COVID-19 case. Staff were also told not to come to work sick because of the risk of COVID-19 transmission, and cover was made available for all absences due to illness. This stance is unusual in Irish hospitals, where doctors are often under pressure to attend work while ill [11]. These policies enabled doctors to take sick leave and time away from work to recover their health (e.g., Shannon, Sean). Workplace-acquired infections are an occupational hazard for health workers worldwide; healthcare workers are reportedly 10 times more likely to return a positive COVID-19 test than the general public [51], and incidences of other highly transmissible infectious diseases such as seasonal influenza, tuberculosis, and avian and swine influenza have previously also been found to be high among healthcare workers [52]. Of the healthcare workers who contracted COVID-19 in Ireland’s first wave, 88% were found to have acquired the virus through work [21]. Yet doctors in Ireland often work through sickness (presenteeism) rather than taking time off work to recover [14]. Strong staff-focused mandatory leave policies, such as those introduced by HSE Occupational Health administrators in the first wave of COVID-19, and increasing staff to cover sick leave, provides time for sick doctors to recover and prevents further transmission to their colleagues and patients. Continuation of these policies beyond the COVID-19 pandemic should be strongly considered to maintain doctors’ wellbeing, and ensure healthy hospital conditions for both doctors and patients.

However, despite doctors having the time to improve their well-being, public health protocols in place to prevent the spread of the pandemic meant that doctors also lost key informal support structures: access to family members and friends, exercise facilities and recreational spaces and events. Some were drawn to coping mechanisms that negatively affected their physical health, like alcohol and overeating, as has been documented among doctors during COVID-19 lockdowns [53]. Others were drawn to use work as a coping mechanism and undertake additional work, like Sean and Bridget, which may cause or reinforce poor work–life balance [15] which may damage doctors’ longer-term well-being and accelerate burnout [6]. While workplace well-being supports are largely focused on supporting doctors in their physical, social and emotional experiences in the workplace, they also need to consider doctors’ total life space, and help doctors acknowledge and address how isolation or family pressures outside the workplace might be affecting them.

### 4.2. The Emotions of Working in the Context of COVID-19

Our findings reveal a broad range of emotions that doctors experienced working during the first wave of COVID-19. Some doctors reported positive emotions: pride in their work, relief and hope as the worst-case scenarios of overwhelming COVID-19 cases did not come to pass, and inspiration when surrounded by their dedicated peers and colleagues. Negative emotions also came across strongly in interviews—fear, guilt, sadness, frustration—similar to those expressed in studies of other frontline staff working in COVID-19 [1,2,3,4,5]. These negative emotions can all build up into emotional exhaustion, which is a major component of burnout [31], a serious concern for health workers during COVID-19, as the World Health Organisation has recently identified [9]. Therefore, it is important for doctors to acknowledge and reflect on emotions at work during the pandemic, as an important part of their wellbeing.

Fear was evident in many doctors’ stories (Fiona), particularly in the early stages of the pandemic on seeing the effects of the virus in other countries. This may have been exacerbated by doctors’ awareness of the shortcomings of the Irish health system, particularly in terms of infectious disease and intensive care hospital capacity [11]. Unclear preparation and communication from hospital administration, as Shannon describes, also caused fear. The specifics of caring for patients with or at risk of COVID-19, especially in full PPE, was fear-inducing for some, like Aidan, and notably more stressful than the everyday practice of medicine [2,8]. Many doctors (Shannon) also voiced fear about the build-up of non-COVID-19 illnesses and workload after the acute phase of the epidemic, a fear mirrored in studies of doctors in other countries [4].

Many doctors (Shannon, Fiona) felt guilty about not being able to provide the usual care to their patients. Unlike other countries, Ireland’s hospitals had sufficient facilities and resources to treat all COVID-19 patients. However, as this was partly achieved by cancelling, reducing or refocusing non-COVID-19 care, doctors were unable to adequately see many long-term patients, or did not see non-emergency cases until they became chronic emergencies. Doctors felt guilt remembering the patients who did not receive the care they needed during the first wave of the pandemic. When they could see patients, the focus on COVID-related procedures and requirements (like PPE and physical distancing) made it difficult to build a rapport or offer comfort, and doctors (like Bridget) felt guilt that they could not care for patients in a way that addressed their emotional needs as well as medical. Contracting COVID-19, although likely as a workplace-acquired infection, made doctors feel guilty about taking time off (Shannon) or about infecting colleagues (Aidan).

Working on wards or away from their training sites through the pandemic was a source of disappointment and sadness for some doctors. They also worried about the impact of these disruptions during COVID-19 on their future professional opportunities in the Irish medical system, which is highly competitive [11,54]. Doctors also felt sad when isolated from their peers due to infection control protocols and redeployment purposes (Fiona), missing the vital support colleagues have been shown to provide to healthcare workers [36,37], and particularly sad in quarantine isolation after testing positive for COVID-19. With the high incidence of COVID-19 infections among health workers [51], such disappointment is likely to be widespread.

These experiences of fear, guilt and sadness all impact doctors’ mental well-being, which in turn can affect physical well-being through lack of sleep and use of poor coping mechanisms [31]. In terms of doctors’ Quality of Work Life [26], their fear reflects their lack of safe and healthy working conditions (especially mentally), their guilt reflects their lack of opportunity to develop and use their skills and capacities, and their sadness reflects their lack of social integration (Table 2). In order to combat negative emotions and foster positive ones, doctors will need to reflect on the emotional aspect of what has been experienced in the first wave of the pandemic, although this will be challenging as they are pressed to move back to focus on coping with the care backlog in addition to subsequent waves of the pandemic.

### 4.3. Lockdown and Life beyond the Hospital

Similar challenges to wellbeing are found in the emotions that doctors experienced in their lives at home during the lockdown. Some doctors reported positive emotions: joy and love due to spending more time with family, and serenity and contentment at a more home-based life. Negative emotions also arose in interviews: fear, sadness and disappointment. Many doctors feared the possibility of spreading COVID-19 to family and friends. While PPE and infection control protocols and resources empower doctors somewhat against this fear at work, these protections do not extend to their home lives. This places a burden on doctors to either create infection control procedures at home with their family (like Fiona and Sean) or to isolate themselves from the family for their protection (like Shannon with her children and Fiona with her parents). With the heightened rates of healthcare worker COVID-19 infection [51], doctors’ close contacts and family members may be more likely to contract the virus from them. Ireland’s hospital doctors’ experiences in this regard mirror the international experiences of doctors regarding the fear of transmitting the virus [1,2,4,5,6].

Doctors felt disappointment and sadness when unable to be with family and friends. Some felt compelled to do this for themselves, arranging for others to care for their children outside the home (Shannon) or to make required care visits to their elderly parents (Fiona), just as they went to extreme measures to protect their families in the home. Others spoke about being shunned by family and friends (Aidan, Sean). Similar stigma from the public labelling of healthcare workers as vectors for the virus has been reported elsewhere [55] and although the physical threat of violence to doctors seen in other countries was not evident in Ireland, stigma still poses a threat to mental well-being [2].

Pandemic-related travel restrictions also left some doctors isolated from family support, as Aidan experienced. Regular geographical movement (known as rotation) for junior doctors is an integral part of medical training in Ireland [54], but those who were isolated geographically before the pandemic were even further isolated when pandemic-related travel restrictions impeded their ability to travel elsewhere in Ireland to visit family and friends. Social isolation, whether geographical or due to others’ fears of the virus, can have severe effects on mental and physical health [56], but may go unnoticed while doctors are at work, so support initiatives will need to link in with doctors to gauge their individual circumstances.

Recognition of the total life space is key in Quality of Work Life [26], and family and friends offer important informal well-being support [35]. All doctors in the study had individual, complex priorities and experiences at home, caring for their families, protecting them from the virus and trying to connect with them while also complying with public health restrictions. There is significant emotional spill-over between the home and work spaces [27], so supporting doctors at work also requires asking about, listening to and acknowledging their challenges and support needs at home.

### 4.4. Supporting and Protecting Hospital Doctors

The themes and findings arising from these narratives have great potential implications for policy and practice, both at an organisational (hospital) and system level. Doctors who worked during the COVID-19 pandemic will require support before, during and after the immediate challenge [31]. They need time to restore their mental and physical well-being, space (physical and figurative) to undertake these activities, and recognition of the physical and emotional toll of their experiences. In the fight against COVID-19, doctors and other healthcare workers *“will be the heroes of the day, but we will need them for tomorrow”* [57]; thus, doctors’ well-being must be properly supported to prevent burnout from the long-term effects of pandemic experiences. Research has found that medical professionals generally tend to be slow or reluctant to seek support themselves [7,31,32], but avoiding speaking about experiences and emotions brings a high risk of burnout [57]. Therefore, health systems and hospitals need to proactively ask doctors about their wellbeing, make the time and space to actively listen, and offer appropriate supports as needed by each individual. Four key facets of the QWL framework [26] in particular need to be addressed to support doctors’ well-being. Doctors require safe and healthy working conditions (ii), particularly during an infectious disease outbreak; opportunities to develop and use their skills and capacities (iii); social integration in their work (v), particularly considering the work–life imbalance and isolation many doctors experience; and recognition of their total life space and their personal obligations and capabilities at home (vii).

As Gerada argues, “*In this second wave of COVID, medical staff on the frontline will need mental-health support almost as much as face masks*” [58]. Just as health systems and hospitals consider it their responsibility to provide PPE to their doctors to maintain a physically safe and healthy workplace, they should be equally responsible for providing protections for mental and emotional health. Support will need to provide doctors with protected time and space for reflection in which they can acknowledge and discuss the emotional toll of working through the pandemic. These need to follow evidence-based intervention protocols, and be offered regularly through subsequent phases of the pandemic and in its aftermath, rather than one-time sessions which may do more harm than good [57]. Tools and intervention programs such as those in use by the USA’s National Centre for PTSD [8] or the Schwarz Round program in use in many hospitals internationally [38,57], may provide a good evidence-based model for implementation by hospitals for their healthcare workers.

The PPE and social distancing measures that help maintain a physically safe and healthy workplace for doctors can, nevertheless, hamper their efforts to use their skills and capacities to deliver the usual care to patients. Hospitals will need to acknowledge doctors’ frustration and guilt at not being able to do their job as usual in order to support their well-being, and seek new solutions to connect doctors, patients and families to allow for care and rapport. Medical training bodies will need to consider the effects of COVID-19 redeployment and the reduction in medical procedures on trainee doctors’ abilities to develop their professional skills and capacities [59].

Supportive social integration and communication among healthcare co-workers is crucial for combatting burnout and reducing emotional exhaustion; emotional support in the workplace has been found to be more effective for healthcare workers than support outside work [37]. When doctors do not have their usual support networks of friends and family due to the pandemic, work becomes a vital source of socialization and closeness. Hospitals will need to ensure sufficient and suitable time and space is available, with physical safety in mind, for doctors to continue to interact socially with one another.

Hospitals must also recognize the total life space of doctors and the pressures the pandemic imposes on life outside work, and help ensure time and space is provided for doctors to protect those they are responsible for at home. Many doctors have parenting responsibilities, or live with or care for other family members (e.g., parents, grandparents) who are at increased risk of infectious diseases like COVID-19. Unlike in other countries, the Irish government did not arrange childcare or schooling support for healthcare workers [60], leaving doctors to make often difficult decisions and sacrifices to be able to meet their professional responsibilities. Recognition needs to be made of doctors’ caregiving responsibilities at home, especially during periods of broad societal restrictions, and hospitals need to enable doctors to contribute their professional capabilities to fight COVID-19 without these compromising home responsibilities.

In the course of adapting work practices to meet the demands of COVID-19, many measures taken by hospitals have highlighted (and in some cases, addressed) existing problems within the health service which have long challenged doctors’ quality of working life; chief among them staffing, sick leave and improved infection control measures. Having introduced such measures due to COVID-19, hospitals should strongly consider maintaining and evaluating such measures longer-term to ensure the higher quality patient care they facilitate can be maintained in the post-COVID health service.

## 5. Limitations and Further Research

This study provides a rich insight into the lived experiences of hospital doctors during the COVID-19 pandemic, in their own words, and how their wellbeing and quality of working life has been affected. However, several limitations are evident within this study and its findings. Foremost, the nature of qualitative research means that studies do not aim to provide generalizable results, rather to highlight common themes and issues. While the narratives presented in this paper aim to highlight common themes across the experiences of many doctors, and how these connect to broader known issues and models, they cannot (and do not aim to) represent the lived experiences of all hospital doctors in Ireland or internationally. Furthermore, while this study focuses on the experiences of hospital doctors, the health service involves a wide range of other staff, in both hospital and primary or community care settings. Further research involving other cohorts of health workers within the Irish system would build a more holistic picture of the effects of the COVID-19 pandemic on the health sector. Additionally, these findings were unable to capture the specific experiences of international medical graduates (IMGs) working as hospital doctors in Ireland. Although over 40% of doctors registered to practice in Ireland are IMGs [61], none answered the call for interview participants. Further research involving a targeted study of the experiences of these doctors during COVID-19 would be a welcome contribution to existing literature on the experiences of IMGs in the Irish healthcare system [54,62]. Finally, while this study touches on some of the identified short-medium term impacts that narrowed clinical exposure will have on the career development of non-consultant hospital doctors (NCHDs) in training, it is unable to predict the impact that experiences during COVID-19 may have on their future career decisions. Follow-up research with the cohort of NCHDs in this study as they approach training and career milestones and choices in the next 12–24 months could prove useful for both health workforce scholarship and local workforce planning and training initiatives.

## 6. Conclusions

The stories of these Irish hospital doctors show that well-being at work and at home has been greatly impacted by the COVID-19 pandemic and related restrictions. While the findings apply in the case of hospital doctors in Ireland only, which may limit their generalisability, there is strong correspondence with international findings worldwide, and they may offer a basis for further comparative research. There may be a limitation that the findings are retrospective; however, reflections over time can be useful in capturing initial reactions and adjustment to the crisis, and may offer the opportunity for a future longitudinal follow-up study.

The doctors in this study shoulder a double burden in events like COVID-19, facing the same societal changes and emotional stressors as everyone, alongside greater risk of exposure and additional work pressures. Their risk of COVID-19 also affected these doctors’ families, causing further anxiety. Existing stresses over health system strain, burnout and work–life balance were exacerbated by the pandemic. However, there were few suitable supports in place that proactively gave these doctors the space and time to acknowledge, discuss and address the practical and emotional toll of working through the pandemic. In order to prevent burnout and allow them to continue the vital provision of patient care as the COVID-19 pandemic continues, it is vital that such supports be put in place for hospital doctors and other healthcare workers.

## Figures and Tables

**Table 1 ijerph-18-02051-t001:** Interviewee Profile (*n* = 48).

Gender	
Male	11
Female	37
Specialty Area	
Pre-Specialty	5
Emergency and General Internal Medicine	11
Specialist Internal Medicine	12
Psychiatry	2
Surgery	3
Other	11
Professional Grade	
Intern ^1^	6
Senior House Officer ^2^	13
Registrar ^3^	3
Specialist Registrar ^4^	8
Consultant ^5^	18
Caring Responsibilities ^6^	
Yes	17
No	31

^1^ New graduate doctor; ^2^ Basic specialty trainee (or equivalent) doctor; ^3^ higher specialty (trainee or equivalent) doctor; ^4^ advanced higher specialty trainee doctor; ^5^ fully trained specialist; ^6^ e.g., children, elderly parents.

**Table 2 ijerph-18-02051-t002:** Key elements of the Quality of Life (QWL) framework and alignment with study findings and experiences during COVID-19.

QWL Framework Category	Findings from Narratives	Discussion Themes
ii—safe and healthy working conditions	Early fear of virus; fear of future waves and backlog; Working in Personal Protective Equipment (PPE); breaks/leave	Time and space for well-being; the emotions of working in COVID-19 (fear)
iii—opportunity to develop and use skills and capacities	Inability to provide regular care; PPE and patient rapport	The emotions of working in COVID-19 (guilt)
v—social integration	Ward-based work; redeployment; missing training; isolation	Time and space for well-being; the emotions of working in COVID-19 (sadness)
vii—recognition of total life space	Fear of transmitting virus to family; childcare/elder care; stigma; social isolation	Time and space for well-being; life beyond lockdown

## Data Availability

Data sharing is not applicable to this article.

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
