# Peer review of "“We All Really Need to just Take a Breath”: Composite Narratives of Hospital Doctors’ Well-Being during the COVID-19 Pandemic"

_ijerph, 2021, doi:10.3390/ijerph18042051_

Round 1
Reviewer 1 Report
This is a highly relevant, and well written paper that qualitatively explores the complex interactions between the COVID-19 pandemic, the Irish health system, and doctors wellbeing. The exploration and discussion is both sophisticated and nuanced.
I have the following comments:
- I would like to know more about the 48 doctors samples. In particular, how many were working in hospital vs. community settings pre-COVID, and did any continue providing community-based care during the pandemic? Did all doctors work directly in treating COVID during the pandemic, or did some continue to provide non-COVID related care (other than being reassigned due to distress)? Were any of the doctors of a mental health background? Were any doctors born/trained/have family overseas?
- The authors rightly explore the short-medium term impact that narrowed clinical exposure will have on the junior medical staff, particularly in cementing their clinical skills and knowledge base. However, I would like to see at least some comment on how early clinical experiences can influence the decisions junior doctors make about the kind of medicine they intend to practice in their career. It remains unclear how these decisions will be affected by the repurposing of junior staff in response to the pandemic.
- It is clear from the narratives that there has been a general increase in stressors (threat of infection, uncertainty, increased workloads, adjusting to multiple changes both professionally and socially) while facing diminished access to their usual coping strategies (unable to access friends, family, healthy habits). The authors are correct to identify how this might increase staff dependence on hospital initiatives for wellbeing. Are there successful examples from international literature the authors could cite as concrete suggestions?
- Staff and hospitals also appear to have utilised short term coping strategies ("running on adrenaline", increasing staff hours) that resulted in a pattern of early raised wellbeing that diminished as the pandemic outlasted the longevity of the coping strategy. This lines up with some models from previous pandemics.
- Some of the service-level measures taken to address the pandemic appear to have highlighted (and in some cases addressed) existing problems in the service. The authors rightly draw attention to this, but it is a unique opportunity to improve services post-COVID, and should be worth mentioning that some measures shouldn't be rolled back, and is a topic to be explored further in other research.
- I am surprised that community reaction to the efforts of healthcare staff did not arise in the narratives, either positively or negatively. It would provide some context to the narratives to mention in the introduction/background how the Irish public responded to healthcare staff efforts. Internationally there have been reports of strong community support in some areas (such as applauding the healthcare staff) stigma in others (such as treating healthcare staff as "unclean"), or minimising the seriousness of the pandemic (and thus undermining the efforts of healthcare staff).
- It is worth mentioning that while this paper focuses on the wellbeing of doctors, that the health service involves a wide range of other staff (nursing, allied health, administrative etc) who will be equally affected by COVID-19 and that they work together in an integrated team. While it's outside the scope of this paper to explore their experience, it is certainly an area worth further exploration.
Thank you for the opportunity to review this paper.
Author Response
Dear reviewer, thank you for your generous comments and insightful suggestions on our paper. We have revised the paper to address your comments as follows:
“I would like to know more about the 48 doctors samples” – We have expanded Table 1 in the paper to incorporate doctors’ specialty areas, though we have presented this as aggregate data to prevent identification. As all our respondent doctors were hospital-based rather than community-based, and all were Irish citizens rather than IMGs (a limitation we have added to a new Limitations and Further Research section), we have amended to reflect this in the text. We have also amended to reflect in the text where some specialties worked directly with patients with COVID-19, and others worked on non-COVID care but had their work practices and conditions impacted by COVID restrictions and policies.
“I would like to see at least some comment on how early clinical experiences can influence the decisions junior doctors make about the kind of medicine they intend to practice in their career” – With respect, this was not a focus of our research and was not addressed in interviews, and all we could offer on this topic would be pure speculation. However, it would prove an excellent question for future research and could potentially pose a follow-up study with the non-consultant hospital doctors in this study as they approach training and career milestones and choices in the next 12-24 months; this has been added to a new Limitations and Future Research section.
“….hospital initiatives for wellbeing. Are there successful examples from international literature the authors could cite as concrete suggestions?” – Additional citations to successful initiatives of this nature have been added to the Discussion section (specifically section 4.4).
“worth mentioning that some measures shouldn't be rolled back, and is a topic to be explored further in other research” - This has also been added to conclude the Discussion section.
“community reaction to the efforts of healthcare staff did not arise in the narratives…It would provide some context to the narratives to mention in the introduction/background how the Irish public responded to healthcare staff efforts” – Some data on this theme was found in interviews but removed from initial drafts of this paper due to the complex nature of thinking around public perceptions and the “healthcare hero” narrative; we found incorporating this in both findings and discussions was overburdening the paper and detracting from the core narrative of necessary supports. We felt that the intricacies of the health worker’s hero-vector complex, public perceptions and tropes of health workers, and health workers’ own perceptions of their place in the efforts against COVID-19 needed addressing in a separate paper in its own right, which we currently are in the process of working on. Nevertheless, in the introduction, we have added a brief outline of some of the Irish public reactions to frontline worker efforts in the first wave of COVID-19.
“While this paper focuses on the wellbeing of doctors, that the health service involves a wide range of other staff” – research for this paper came from the HRB funded Hospital Doctor Retention and Motivation (HDRM) Project which is why the focus of the paper is restricted to hospital doctors. However, the point that a wide range of healthcare workers are similarly affected is certainly true, and the introduction and conclusion have been amended to reflect this.
Reviewer 2 Report
The manuscript is well designed and gives us a clearly inside of doctors wellbeing in hospitals during the first pandemic wave. The only identified lack is the unclear methodological articulation. Calling it a thematic analysis is not enough and should be revised. For example analysing by using Mayring`s content analysing approach would be appropiate.
Author Response
Dear reviewer, thank you for your generous comments on our paper.
With regards to your major suggestion, “The only identified lack is the unclear methodological articulation. Calling it a thematic analysis is not enough and should be revised”, we accept the need to further articulate the style of TA used and the approach taken, and have expanded and amended the methods section to highlight our use of the Framework method of structured thematic analysis (Gale et al 2013) on the initial overall dataset to narrow down to wellbeing-related experiences as defined by the Quality of Work Life framework used throughout the paper.
Reviewer 3 Report
Thank you very much for inviting me to review this interesting article. It uses a rather unusual way to present results, which rises some questions. However, wellbeing of doctors (and also other health professionals) during the pandemic is an important topic. Additionally, the article is a good contemporary witness. It is based on a rich amount of recent literature. It focuses very (too) much on doctors. At some points, especially in the introduction and the discussion, the perspective could be broadened or at least linguistically, synonyms for ‘doctor’ could be taken into consideration.
Introduction
The introduction is based on a huge amount of mostly very recent evidence. This is in the nature of the current situation with many new articles about the challenges during the Covid-19 pandemic. The structure of the introduction is rather unusual and there are some concerns:
- The introduction is very, very long.
- Lines 35-39: it takes a little long until the first reference appears.
- Lines 54-58: it is unusual to provide aim and methodological details in the middle of the introduction. Consequently, the end of the introduction does not lead to the rational of the study.
Methods
The structure of the methods section rises also some concerns:
- Lines 154-155: The ethical statement is correct and important but interrupts the flow. It might be better placed in the end of a paragraph.
- Table 1 would belong at the beginning of the result section.
- A rational for presenting the results in narrative interviews is provided. It seems that it was more important to provide a good contemporary witness and address politicians than to present results scientifically. However, it is difficult to understand, why data was coded. How did the codes and themes play a role in the development of the narratives?
Results
The problem of combining stories is, that some aspects lost their logic:
- Sean was fast-tracked into an Intern post at the end of Medical School. On lines 225-226 however it is written, that his usual public transport commute is hour-long. He only started work during the pandemic. How can there be a change in the commute?
- Shannon was the first to declare that her physical health declined. However, on line 275 it is written, that her health “also” declined.
- Shannon is working on a busy Covid-19 ward. However, on line 283 it is mentioned that she had a reduced workload. And on line 287 she speaks about the admission to the hospital for other reasons. Is she working on different wards?
- Fiona is sitting alone in her office and provides telemedicine. Why did she stop to visit her parent because of a risk to transmit the Covid-19? Did she have other risk factors to catch the disease than her work? Additionally, she organised meals for her team despite working alone in her office…
Discussion
The discussion is based on an interesting frame. The author should consider a few changes:
- Table 2: It is not clear, what the headings ‘Title 2’ and ‘Title 3’ mean.
- Lines 547 and following: There are too many ‘doctors’ plus verb. Language in this part should be revised to avoid boring sentences.
- It would be meaningful to provide a clear paragraph about strengths and limitations instead of only mention a few limitations in the conclusion.
Conclusion
- Qualitative studies do not aim to provide results, which can be generalised. Is there a need to address this as a potential limitation?
Author Response
Dear reviewer, thank you for your thorough and insightful comments, which have highlighted for us a particular need to reinforce the validity of our narrative methods, which are in common use in the disciplinary areas of some of our authors but which we understand now are out of the ordinary in other disciplines. With regards to your specific suggestions:
“It focuses very (too) much on doctors” – While we recognize that healthcare is delivered by a range of health workers and that all are experiencing wellbeing issues in this COVID-19 pandemic, research for this paper came from the HRB funded Hospital Doctor Retention and Motivation (HDRM) Project which is why the focus of the paper is restricted to hospital doctors. We have amended the introduction and conclusion to recognize that health systems need to protect the wellbeing of all healthcare workers, not just doctors. We have used the term “doctors” throughout rather than alternating with “physicians” as this is the terminology the Irish Health Service Executive uses to describe hospital doctors in particular.
“The introduction is very, very long... it is unusual to provide aim and methodological details in the middle of the introduction. Consequently, the end of the introduction does not lead to the rationale of the study.” – We have amended this by moving the aim and methodological details (previously lines 54-59) to the end of the introduction after the wellbeing literature review (now lines 147-152).
“Lines 154-155: The ethical statement is correct and important but interrupts the flow. It might be better placed at the end of a paragraph” – we have moved this as suggested.
“Table 1 would belong at the beginning of the result section” – other reviewers have requested that Table 1 be expanded and more provided in the Methods section about who was approached for interviews and who interviewees were; moving this expanded table to the results section would take it out of the context of this expanded description.
“It seems that it was more important to provide a good contemporary witness and address politicians than to present results scientifically” – we would respectfully submit that storytelling is key in applied social science, like this study, for translating evidence into policy. (See for example Davidson B. Storytelling and evidence-based policy: lessons from the grey literature. Palgrave Communications 2017;3:17093.) The policy impact we are hoping to achieve is to ensure that the wellbeing of frontline hospital staff is central to the ongoing pandemic response and the post-pandemic recovery. The narratives, or “witness” as you term it, were designed to simply, quickly, and effectively summarize the key issues and concepts regarding wellbeing that respondents identified within the complexities of individual lives, telling a broader thematic story with a human face to convince key decision-makers of the policy effects on lives. We have edited our description of our narrative method to highlight our aims better in this regard.
“why data was coded. How did the codes and themes play a role in the development of the narratives?” – we have expanded and amended the methods section to highlight our use of the Framework Method of structured thematic analysis (Gale et al 2013) on the initial overall dataset to narrow down to wellbeing-related experiences as defined by the Quality of Work Life framework used throughout the paper (lines 178-182). We have also expanded (lines 197-200) on the progression between coded data and crafted narratives.
“The problem of combining stories is, that some aspects lost their logic” – we are grateful you have identified these, all have been remedied/further explained in the text.
“Table 2: It is not clear, what the headings ‘Title 2’ and ‘Title 3’ mean” – this formatting error has now been corrected to “Findings from Narratives” and “Discussion Themes” respectively
“Lines 547 and following: There are too many ‘doctors’ plus a verb. Language in this part should be revised” – we have revised some of these to “medical professionals”, a generic “they”, etc.
“provide a clear paragraph about strengths and limitations instead of only mention a few limitations in the conclusion” – an expanded ‘Limitations and further research’ section has been added. It also includes your final suggestion, regarding the aims of qualitative results and the inability to generalize these across the broader medical profession.
Reviewer 4 Report
Dear Authors:
Your manuscript needs to be improved in several respects.
- The theoretical contributions and practical implications of this paper are not clear enough. I have doubts about the necessity of this study. You mainly discussed the composite narratives of hospital doctors’ well-being during the COVID-19 Pandemic, but didn’t explicitly propose specific measures about how to address this problem.
- Introduction: The consequences of poor workplace well-being are very diverse. Why did the authors choose burnout to make a discussion rather than other outcomes? It still needs further clarification.
- The classification of well-being in the article is not clear, and it involves workplace well-being, well-being at home, physical well-being and mental well-being for many times, which may confuse the reader.
- Materials and Methods: (1) The typesetting of Table 1 is not standard. (2) Your manuscript does not list specific interview outline, and the authors are suggested to supplement it as an appendix. (3) The thematic data analysis is not clear enough, which may confuse the readers. It is necessary to indicate more details about the data processing.
- There are some errors in the format and tense of the article, such as the wrong tense (line 20) and inconsistent format (line 194) , so I suggest the authors check it carefully.
Author Response
Dear reviewer, thank you for your thorough and insightful comments, which have highlighted for us a particular need to reinforce the implications and contributions we aimed to make in this paper. With regards to your specific suggestions:
“You mainly discussed the composite narratives of hospital doctors’ well-being during the COVID-19 Pandemic, but didn’t explicitly propose specific measures about how to address this problem” – Our aim in this paper was to highlight issues and needs, rather than make specific policy recommendations, as we have found the latter makes it more difficult to achieve the policy impact we are hoping for (in this case, to ensure that the wellbeing of frontline hospital staff is central to the ongoing pandemic response and the post-pandemic recovery). Nevertheless, we feel the themes and findings arising from these narratives do have great potential implication for policy, both at an organizational (hospital) and system level; we have amended the discussion (specifically section 4.4) to better highlight this, and have explicitly stated that the wellbeing of the medical workforce needs more attention, both in practice and in scholarship.
“Why did the authors choose burnout to make a discussion rather than other outcomes?” – While there are multiple outcomes for poor wellbeing, burnout is, we would respectfully argue, the most prevalent and the most problematic in healthcare, as burnout rates amongst medical professionals are consistently found to be high (and higher than in the general population) in studies internationally. We have amended our literature review section to list additional outcomes for poor wellbeing and explain our focus on burnout.
“The classification of well-being in the article is not clear, and it involves workplace well-being, well-being at home, physical well-being and mental well-being for many times, which may confuse the reader.” – The literature we draw on has multiple classifications and conceptualizations of wellbeing, which do encompass workplace, home, physical and mental wellbeing simultaneously. Our classification of wellbeing is drawn directly from Walton’s Quality of Work Life framework, which we have applied consistently throughout this paper. To help make this classification clearer we have moved our definition and outline of the QWL framework to a separate paragraph on its own, which hopefully reduces confusion.
“The typesetting of Table 1 is not standard” – this has been amended to align better with the journal template and more clearly display data
“Your manuscript does not list specific interview outline, and the authors are suggested to supplement it as an appendix” – This paper is one of a series exploring different themes of a broader overarching semi-structured interview study. While this paper is focused on the impact of COVID-19 on their wellbeing, interviews also discussed their experiences of working as a doctor pre-pandemic, during the pandemic, and their plans for the future (and we have expanded to explain this). Only one interview question among the seven themed questions is relevant to the results of this particular paper – this question, ‘how has the pandemic impacted on your well-being?’, is reproduced in full within the text of the Methods section (line 181). We would respectfully submit that appending the full interview outline would not provide any relevant additional context to this paper.
“The thematic data analysis is not clear enough, which may confuse the readers. It is necessary to indicate more details about the data processing” - we have expanded and amended the methods section to highlight our use of the Framework Method of structured thematic analysis (Gale et al 2013) on the initial overall dataset to narrow down to wellbeing-related experiences as defined by the Quality of Work Life framework used throughout the paper (lines 178-182). We have also expanded (lines 197-200) on the progression between coded data and crafted narratives.
“There are some errors in the format and tense of the article, such as the wrong tense (line 20) and inconsistent format (line 194), so I suggest the authors check it carefully” – thank you for drawing this to our attention; we have had the text of the paper thoroughly edited for US spelling variants, grammar, and expression.